# Metagenomic detection of eumycetoma causative agents   from households of patients residing in two Sudanese endemic villages in White Nile State

Antonella Santona[1], Najwa A. Mhmoud[2,3], Emmanuel Edwar Siddig [2,3], Massimo Deligios[1], Maura Fiamma[1], Bianca Paglietti[1], Sahar Mubarak Bakhiet[2,4], Salvatore Rubino[1], Ahmed Hassan Fahal[2]*

1 Department of Biomedical Sciences, University of Sassari, Sassari, Italy, 2 Mycetoma Research Centre, University of Khartoum, Khartoum, Sudan, 3 Faculty of Medical Laboratory Sciences, University of Khartoum, Khartoum, Sudan, 4 Institute for Endemic Diseases, University of Khartoum, Khartoum, Sudan

* ahfahal@hotmail.com, ahfahal@mycetoma.edu.sd

**Data Availability Statement:** All relevant data are within the manuscript and its Supporting Information files. Furthermore, all raw sequencing

## Abstract

Eumycetoma is a chronic debilitating fungal disease endemic to tropical and subtropical regions, with Sudan featuring the highest eumycetoma incidence. Among the 50 species of fungi most commonly associated with eumycetoma *Madurella mycetomatis* (*M. mycetomatis*) is often referenced as the most common pathogen. However, there is an enormous knowledge gap related to this neglected disease and its pathogenesis, epidemiological features, and host-specific factors that could contribute to either the host susceptibility and resistance. In this study, we were able to utilize a metagenomic approach and samples collected from clinical black grains (BG) and familiar household environments aimed to assay both the habitat of eumycetoma-associated fungi and its possible connection with eumycetoma patients living in two different eumycetoma endemic villages within the White Nile State of Sudan. DNA sequencing targeting the fungal ITS2 domain was performed on soil, animal dung, housing walls and roofs, and *Acacia*-species thorn samples and compared with culture-dependent methods of fungal isolation. Additionally, we compared the soil samples obtained in the endemic zone with that from non-endemic zones, including Wagga village in Kassala State and Port Sudan suburb in Port Sudan State. Overall, a total of 392 Amplicon Sequence Variants (ASVs) were detected by ITS2 metagenomics Eumycetoma causative organisms accounted for 10% of total ASVs which included *11 genera*: *Exserohilum* (2%), *Aspergillus (1.7%)*, *Curvularia* (1%), *Alternaria* (0.9%), *Madurella* (0.5%), *Fusarium* (0.4%), *Cladosporium* (0.2%) *Exophiala* (0.15%), *and, in a lesser extent*, *Microascus* (0.05%) *Bipolaris* and *Acremonium* (0.01%) for each. Only five genera were identified by culture method, which included *Fusarium* (29%), *Aspergillus* (28%), *Alternaria* (2.5%), *Bipolaris* (1.6%), and *Chaetomium* (0.8%). *M. mycetomatis* was detected within all the studied patients' houses, accounting for 0.7% of total sequences. It was the first common eumycetoma-associated agent detected in soil samples and the third common in the dung and wall samples. In contrast, it was not detected in the roof or thorn samples nor in the soils from non-endemic regions. *Exserohilum rostratum*, *Aspergillus spp* and *Cladosporium spp* were

data generated in this study are accessible in the NCBI database under the ID PRJNA732838.

**Funding:** AS was supported by the Italian Agency for Development Cooperation (AICS) Grants AID 10821 and AID10861. The funders had no role in study design, data collection and analysis, decision to publish, or preparation of the manuscript.

**Competing interests:** The authors have declared that no competing interests exist.

detected in all samples. *M. mycetomatis* and other eumycetoma-associated fungal identified in the patients' black grains (BG) samples by metagenomics were identified in the environmental samples. Only *Acremonium alternatum* and *Falciformispora senegalensis*, responsible for eumycetoma in two patients were not detected, suggesting the infections in these patients happened outside these endemic areas. The soil, animal dung, and houses built from the same soil and dung are the main risk factors for *M. mycetomatis* infection in these endemic villages. Furthermore, the poor hygienic and environmental conditions, walking barefooted, and the presence of animals within the houses increase the risk of *M. mycetomatis* and other fungi causing eumycetoma.

## Author summary

In this pilot study, using a metagenomic approach, we revealed in two Sudanese eumycetoma endemic villages within the While Nile State in Sudan, the habitat of *M. mycetomatis* and other fungal species responsible for eumycetoma. Although never isolated in culture, *M. mycetomatis* represented the most abundant eumycetoma-associated species found within soil samples and the third most common species within dung and housing wall samples. All the eumycetoma-associated fungal species detected by metagenomic in black grains samples were identified in patient's houses, except *Falciformispora senegalensis* and *Acremonium alternatum*. The findings obtained in this study provided insight into the habitat of eumycetoma-associated causative species and improved knowledge on eumycetoma origin and risk factors in endemic villages. Furthermore, despite the limited number of samples, these results suggest the main prevention measurements to contain eumycetoma in these endemic areas. These measurements include using gloves and alternative materials to endemic soil and animal dung in building the wall of the houses walls, constructing animal fences and appropriate use of footwear.

## Introduction

Mycetoma is a chronic subcutaneous granulomatous, inflammatory and debilitating neglected disease [1]. It is classified into eumycetoma and actinomycetoma, depending on whether the causative agent is fungal or actinobacteria, respectively. Between the two diseases, eumycetoma is more common. In Sudan it accounts for an estimated 70% of all mycetoma cases. [2,3]. Mycetoma is generally endemic to tropical and subtropical climates, particularly in areas featuring moderate aridity, low humidity, and a short rainy season. [4,5].

A recent study found that aridity, proximity to water sources, lower soil concentrations of sodium and calcium, and the distribution of thorn-bearing trees served as the strongest predictors of mycetoma in Sudan [2,6]. The statistical models in that study predicted the highest occurrence of eumycetoma and actinomycetoma in the central and southeastern states of Sudan and along the Nile River valley and its tributaries [2].

In Sudan, the infection is commonly seen in the feet and hands [2]. It frequently affects young adults, mostly males aged 15 to 30, of low socioeconomic status. Shepherds, manual workers, and students are most affected. In general, it affects the poorest of the poor in poor and remote communities [7].

A variety of fungal species, from eight different orders, can cause eumycetoma. Worldwide the primary causative agent is *Madurella mycetomatis* (*M. mycetomatis*) from the order

*Sordariales.* This organism is commonly associated with the production of black grains (BG) within affected tissues. These grains have a cement matrix consisting of melanin, heavy metals, proteins, and lipids that contribute to the organism's pathogenicity impeding the penetration of various antifungal agents [8,9]. Other fungi associated with BG eumycetoma are commonly from the order *Pleosporales*, including *Falciformispora senegalensis* (*F. senegalensis*), *Falciformispora tompkinsii* and *Medicopsis romeroi* [10]. In contrast, pale grain eumycetoma species are typically from the order *Hypocreales*, including *Acremonium spp*, *Cylindrocarpon spp*, and *Fusarium spp* or the orders *Eurotiales* or *Microascales* including *Aspergillus spp* and *Microascus gracilis* respectively [11].

In identifying of eumycetoma causative species, grain culture is the gold standard method, but it has some limitations that are partially overcome by molecular-based techniques. A metagenomic analysis has recently improved their identification even in samples with negative cultures, providing insights into the diversity and complexity of fungal grains and diseases.

Eumycetoma generally responds poorly to the few currently available antifungal therapies, all of which can be toxic, expensive, or generally unavailable within many eumycetoma endemic regions. It may be unsurprising that aggressive surgery up to and including amputation is often the only recourse for treating this disease in those areas [12,13]. This difficulty can likely, at least in part, be attributed to a lack of clarity regarding eumycetoma pathogenesis, epidemiology, and the specific host factors that might contribute to different patients outcome [14].

It is hypothesized that the aetiologic agents are traumatically introduced via thorns-from trees and shrubs of the *Acacia* genus and/or with soil particles into the subcutaneous tissue [15]. Furthermore, it has also been hypothesized that mammal dung, ubiquitously present in rural villages, could play a significant role in the ecology of *Madurella* and *Sordariales* [16]. More recently, ticks have also been proposed as possible *Madurella* eumycetoma risk factors [17]. However, the definitive source of eumycetoma infection has not been yet clarified.

Our study utilizing ITS2 metagenomics demonstrated the presence of multiple fungal species most commonly associated with soil and animal dung, including eumycetoma-associated agents within BG samples obtained from patients living within the eumycetoma endemic region of the White Nile [18]. Hence, we extrapolated that the causative organisms identified in the black grains had a natural environmental habitat, supporting the theory that traumatic subcutaneous environmental material implantation is the most probable route of infection [18].

For this reason, we undertook a similar metagenomic approach to examine soil and household samples from patients diagnosed with eumycetoma to clarify the epidemiology and environmental niche of the fungal "consortia" found within BG samples.

## Materials and methods

### Ethics statement

Ethical approval was obtained from the Mycetoma Research Centre Institutional Review Board (IRB) (MRC2018-01: Approved). For the grain samples a consent form was obtained from the patients.

### Sampling site

Environmental samples were collected from two eumycetoma endemic villages in the White Nile State, Sudan. The villages were El Andalos (N = 5 houses) and Alsobhi (N = 4 houses). The coordinates of each sampling site were determined by GPS (Table 1 and Fig 1).

**Table 1. Patient' houses, villages, sampling sites, kind and number of samples are shown.** Last column shows the eumycetoma diagnosis with the microorganisms previously identified by culture methods or by metagenomics (*).

| House | Village of sampling | Region | Coordinates | Date | No of sample | Soil_S1 | Deep Soil_S2 | Dung_R | Wall_W | Roof | Afood_HRoof | Thorns | BG Metagenomics | Eumycetoma diagnosis |
|---|---|---|---|---|---|---|---|---|---|---|---|---|---|---|
| House 2 | El Andalous | White Nile | 13°45'07.2"N 32°22'56.4"E | 16/11/2017 | 5 | 1 | 1 | 1 | 1 | 1 | | | | *M.m* |
| House 5 | Alsobahi | White Nile | 13°47'27.8"N 32°21'31.4"E | 16/11/2017 | 6 | 1 | 1 | 1 | 1 | 1 | | 1 | 1* | *F.spp/A.a*\* |
| House 7 | El Andalous | White Nile | 13°45'00.8"N 32°23'08.5"E | 16/11/2017 | 6 | 1 | 1 | 1 | 1 | 1 | | 1 | 1* | *M.m*\* |
| House 9 | Alsobahi | White Nile | 13°47'28.3"N 32°21'33.5"E | 16/11/2017 | 4 | 1 | 1 | 1 | | | 1 | | | *C.a* |
| House 10 | Alsobahi | White Nile | 13°47'27.8"N 32°21'31.4"E | 16/11/2017 | 6 | 1 | 1 | 1 | 1 | 1 | | 1 | | *M.m* |
| House 15 | Alsobahi | White Nile | 13°47'28.3"N 32°21'31.0"E | 16/11/2017 | 5 | 1 | 1 | 1 | 1 | | | 1 | 1* | *M.m*\* |
| House 16 | Alsobahi | White Nile | 13°47'30.3"N 32°21'33.7"E | 16/11/2017 | 6 | 1 | 1 | 1 | | 1 | 1 | 1 | | - |
| House 18 | El Andalous | White Nile | 13°45'01.9"N 32°22'52.9"E | 16/11/2017 | 6 | 1 | 1 | | 1 | 1 | | 2 | 1* | *F.s*\* |
| House 21 | El Andalous | White Nile | 13°45'01.8"N 32°22'52.1"E | 16/11/2017 | 5 | 1 | 1 | 1 | 1 | | 1 | | 1* | *M.m*\* |
| | | | | | 49 | 9 | 9 | 8 | 7 | 6 | 3 | 7 | 5 | |
| **Soil from non-endemic villages** | | | | | | | | | | | | | | |
| PS_S1_5 | Port Sudan | Port Sudan | 19°36'06.1"N 37°11'49.4"E | 16/11/2018 | | 1 | | | | | | | | |
| PS_S1_6 | Port Sudan | Port Sudan | 19°36'06.1"N 37°11'49.4"E | 16/11/2018 | | 1 | | | | | | | | |
| K_S1_5I | Wagga | Kassala | 16°09'21.5"N 36°12'36.6"E | 15/11/2018 | | 1 | | | | | | | | |
| K_S1_5 | Wagga | Kassala | 16°09'21.5"N 36°12'36.6"E | 15/11/2018 | | 1 | | | | | | | | |
| K_S1_19 | Wagga | Kassala | 16°09'21.5"N 36°12'36.6"E | 15/11/2018 | | 1 | | | | | | | | |

M. m. = *Madurella mycetomatis*; F. spp = *Fusarium spp*; A. c. = *Acremonium alternatum*, C.a. = *Chaetomium atrobrunneum*; F.s. = *Falciformispora senegalensis*

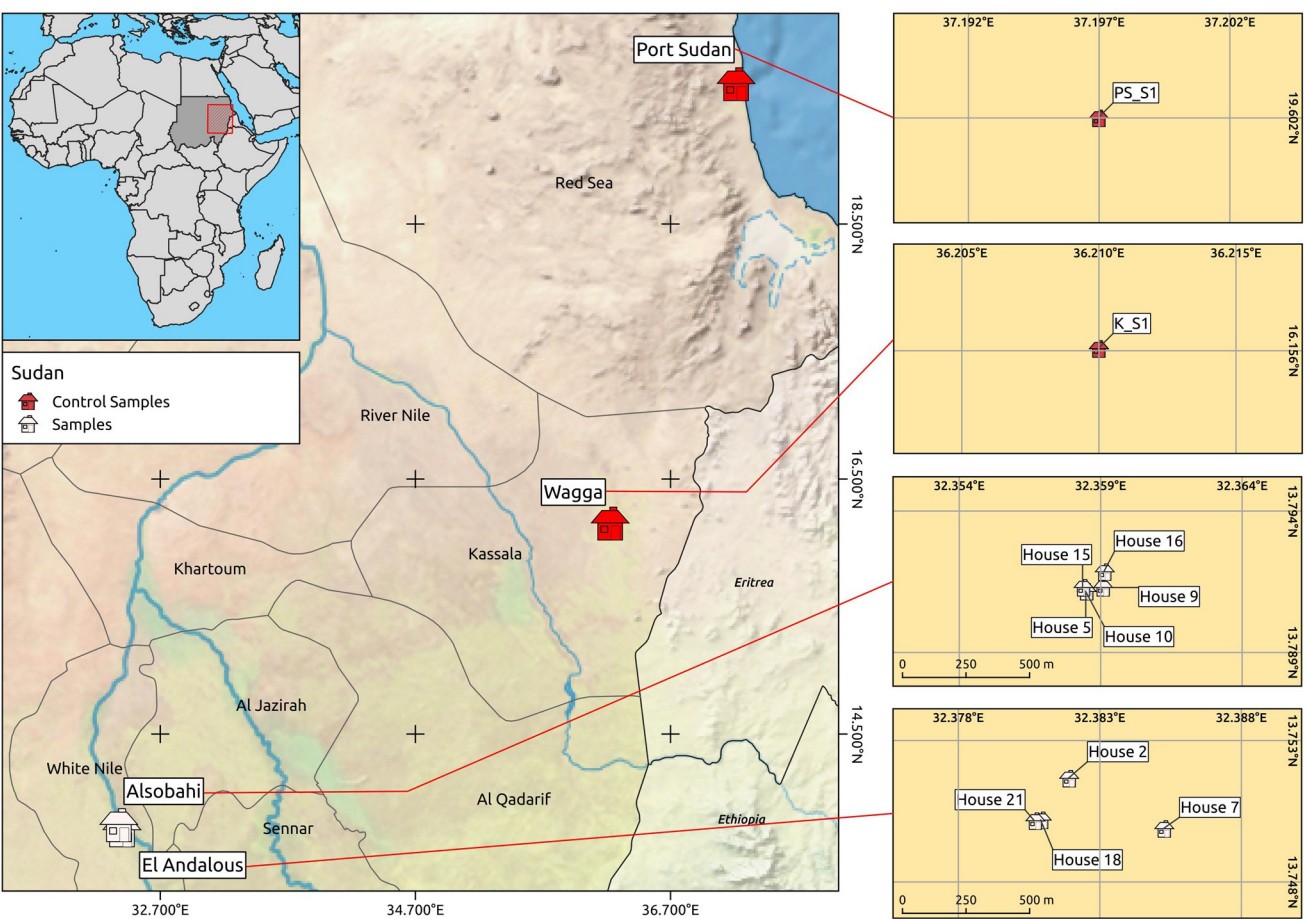

**Fig 1.** Left: sites of sampling from Sudanese eumycetoma endemic (White Nile) and not endemic (Kassala and Port Sudan) regions. Right: images of the sampled houses. The map was generated with QGIS 3.10.4-Coruña on Ubuntu 20.04.4 LTS. Background and administrative limits: http://www. naturalearthdata.com/about/terms-of-use/): https://www.naturalearthdata.com/downloads/10m-cultural-vectors/10m-admin-0-countries/ https://www. naturalearthdata.com/downloads/10m-cultural-vectors/10m-admin-1-states-provinces/ https://www.naturalearthdata.com/downloads/10m-natural-earth-2/10m-natural-earth-2-with-shaded-relief-water-and-drainages/.

In eight houses, there were one or more patients with eumycetoma caused by *M. mycetomatis* (N = 5), *Chaetomium atrobrunneum* (*C. atrobrunneum*) (N = 1), *F. senegalensis* (N = 1) and *Fusarium spp* (N = 1) (Table 1). One house (#16) was without eumycetoma cases. Five houses (# 5, 7, 15, 18, and 21) were from eumycetoma patients previously diagnosed by BG metagenomics [18].

Environmental samples were obtained from superficial soil (N = 9) and soil at 30 cm depth (N = 9), housing samples from walls, mostly made of soil and animal dung (N = 7) and roofs (N = 6), mostly made from local tree/plants, animal dung from domestic cows, goats, and donkeys (N = 8), dried local plants generally fed to livestock (N = 3), and thorns from Acacia species trees local to the area (N = 7) (Table 1).

Additionally, soil samples were collected from non-endemic sites including Wagga village in Kassala State (N = 3) and Port Sudan suburb within Port Sudan State (N = 2) (Table 1 and Fig 1).

The samples were collected utilizing sterile gloves and placed into sterile labeled 50 ml Falcon tubes with a sterile spatula. Samples were then brought to the laboratory to be stored at 4˚C before being portioned for both culture and DNA extraction.

## Culture-dependent methods

For fungi isolation, samples were cultivated in non-selective and selective media, including Columbia blood agar (Microbiol Diagnostici, Italy), Potato dextrose agar (Oxoid, United Kingdom)), Sabouraud dextrose agar with and without chloramphenicol (0.05 g/L) and gentamicin (0.1 g/L) (Microbiol Diagnostici, Italy), and Dermatophyte agar (Microbiol Diagnostici, Italy). Soil and wall samples were circularly plated, as usually done for fungi colonies, while solid samples (dung, roof and thorn) were placed in the center of the plate with light pressure. Plates were incubated at 26°C and 30°C for 5–15 days and checked every two days. Colonies were examined for shape, size, appearance, substrate colour, aerial mycelia, and production of diffusible pigments. The identification of fungi was based on morphological criteria described in Koneman's diagnostic atlas compared with the microscopic morphology of fungi obtained by Lactophenol Cotton Blue stain [19].

Other fungi unable to be identified by culture or morphology were identified by Sanger sequencing. Nucleic acids were extracted from the isolates cultured in Sabouraud liquid media (Oxoid, United Kingdom) using YeaStar Genomic DNA Kit (Zymo Research, Tustin, California; United States). The nucleic acids were amplified by PCR using specific primers for fungi targeting the internal transcribed spacer (ITS2) region of the ribosomal rRNA gene [20]. The specific amplicons were purified using DNA clean and concentrator (DCC) (TM-5-Zymo Research, California, US) and Sanger sequenced (BMR Genomics, Padova). Sequences were analysed using Geneious 11 software (http://www.geneious.com/), and the species were identified by BLAST database sequences comparison (Sheet A in S1 Table).

## Detection of biodiversity within samples by metagenomics

For ITS2 amplicon-based DNA sequencing, 0.25 grams of each sample was aseptically weighed and subjected to total DNA extraction in a biosafety cabinet using DNeasy PowerSoil Kit (Qiagen, Germany) and then resuspended utilizing sterile water (UltraPure Distilled Water, Invitrogen, Carlsbad, California, USA). Contamination between samples was excluded using Ultrapure DNA free water as a separate blank control.

As requested by the sequencing service, samples and controls were checked for ITS2 DNA amplicon sequencing eligibility using specific tailed ITS3 and ITS4 primers for the ITS2 region. The DNA concentration required for metagenomics (range: 3–10 ng/μl) was determined by a Qubit spectrophotometer (Invitrogen, Carlsbad, California, USA).

ITS2 barcode sequencing and Sanger sequencing were performed using the NGS sequencing service offered by the BMR Genomics Company, Padua, Italy. Libraries were prepared using Nextera XT DNA Library with sequencing performed via the Illumina MiSeq Paired-End platform.

Sequences were uploaded on GenBank with the BioProject ID: PRJNA851016.

## Data processing and sequence analysis

Sequences from all samples were joined and imported to Qiime2 package ver. 2018.11 (https://doi.org/10.1038/s41587-019-0209-9). Sequences were filtered, denoised, merged, and subjected to the removal of chimaeras. High-quality sequences were processed to downstream analysis such as taxonomy (in term of ASVs, Amplicon Sequence Variants) and alpha and beta diversity [21]. Regarding the taxonomy, the sequences were compared with the classifier Unite 8.0 clustered at 99% (https://doi.org/10.1093/nar/gky1022). Graphic visualization was made using the package microeco and R environment [https://doi.org/10.1093/femsec/fiaa255]. Alpha diversity was calculated using the Shannon index with samples rarefied at 1600 sequences and with rarefaction curves of the soil samples until 15000 sequences. Beta diversity

was evaluated with unweighted UniFrac and represented as a Principal Coordinates Analysis (PCoA) of all samples and just soil samples.

Meta-data from BG ITS2 metagenomics obtained from five eumycetoma patients with the same DNA extraction procedure and ITS2 base amplicon sequencing was re-analysed with environmental samples using the same settings.

One-way analysis of variance (ANOVA) and Tukey's HSD tests were utilized to determine differences between fungi causing eumycetoma in endemic and non-endemic soil, as well as whether *M. mycetomatis* was more prevalent in the environmental houses of patients affected by *M. mycetomatis* compared to those affected by other species or without cases.

## Results

### Culture dependent method

In this study, 124 fungi were isolated by cultural method from the environmental samples of endemic villages, which included *Fusarium spp* (N = 35), *Aspergillus spp* (N = 35), *Mucor spp* (N = 23), and other species as shown in Sheet A in S1 Table. *M. mycetomatis* was not isolated from any of these samples. Five genera associated with eumycetoma were identified, which included *Fusarium* (28%), *Aspergillus* (28%), *Alternaria* (2.5%), *Bipolaris* (1.6%), and *Chaetomium* (0.8%) (Sheet A in S1 Table). Several species of eumycetoma causative fungi not identified by morphological method were specified by ITS2 Sanger sequencing. These included *Fusarium falciforme* (*F. falciforme*) (N = 1), *Fusarium oxysporum* (N = 1), *Aspergillus flavus* (*A. flavus* (N = 2), *Aspergillus nidulans* (*A. nidulans*) (N = 6), *Aspergillus niger* (*A. niger*) (N = 3), *Chaetomium atrobrunneum* (*C. atrobrunneum*) (N = 1) *and Curvularia spicifera* (N = 1) (Sheet A in S1 Table).

### Fungi identification by culture-independent ITS2 analysis

Overall, 392 ASVs were detected in the environmental samples from the two endemic villages by ITS2 metagenomics. Eumycetoma-associated ASVs (N = 40) accounted for 17% of *Ascomycota* ASVs (N = 239) and 10% of total ASVs (Sheets A and C in S1 Table). *Basidiomycota* represented 25% of ASVs, while 3% of ASVs were from the protistan kingdom *Alveolata*.

As shown by the alpha rarefaction analysis (Fig 2A), samples with the highest ASV diversity were from deep soil, housing walls, and superficial soil. This was followed by animal dung, housing roofs, thorns, and finally, clinical BG samples.

Port Sudan soils had lower ASV diversity than Kassala and White Nile endemic soils (Fig 2B). Houses # 21, 15 and 9 with eumycetoma patients had the highest ASVs diversity, while houses # 18, 7, and 10 had the lowest ASVs diversity (Sheet C in S1 Table).

Beta diversity was calculated as unweighted UniFrac distance (based on the phylogenetic relationships between the ASV) and visualized with a Principal Coordinates Analysis (PCoA) plots (Fig 3). Clinical samples (BGs 5,7,15,18,21) of patients (from houses # 5, 7, 15, 18 and 21) were also included in the plot (Fig 3A).

Compared to total ASV count, ASVs of eumycetoma-associated fungi were found more commonly in housing roof (15%) and wall (13%) samples. This was followed by animal dung samples (11.6%), superficial soil and thorn samples (11.5% each), and deep soil samples (11%) (S1C Table).

ASV metagenomics of samples obtained from the environment of endemic villages identified multiple genera associated with eumycetoma including *Exserohilum* (2%), *Aspergillus* (1.7%), *Curvularia* (1%), *Alternaria* (0.9%), *Madurella* (0.5%), *Fusarium* (0.4%), *Cladosporium* (0.2%), *Exophiala* (0.15%), and to a lesser extent *Microascus* (0.05%) *Bipolaris* and *Acremonium* (0.01%) for each (Sheets B and C in S1 Table).

## ALPHA DIVERSITY

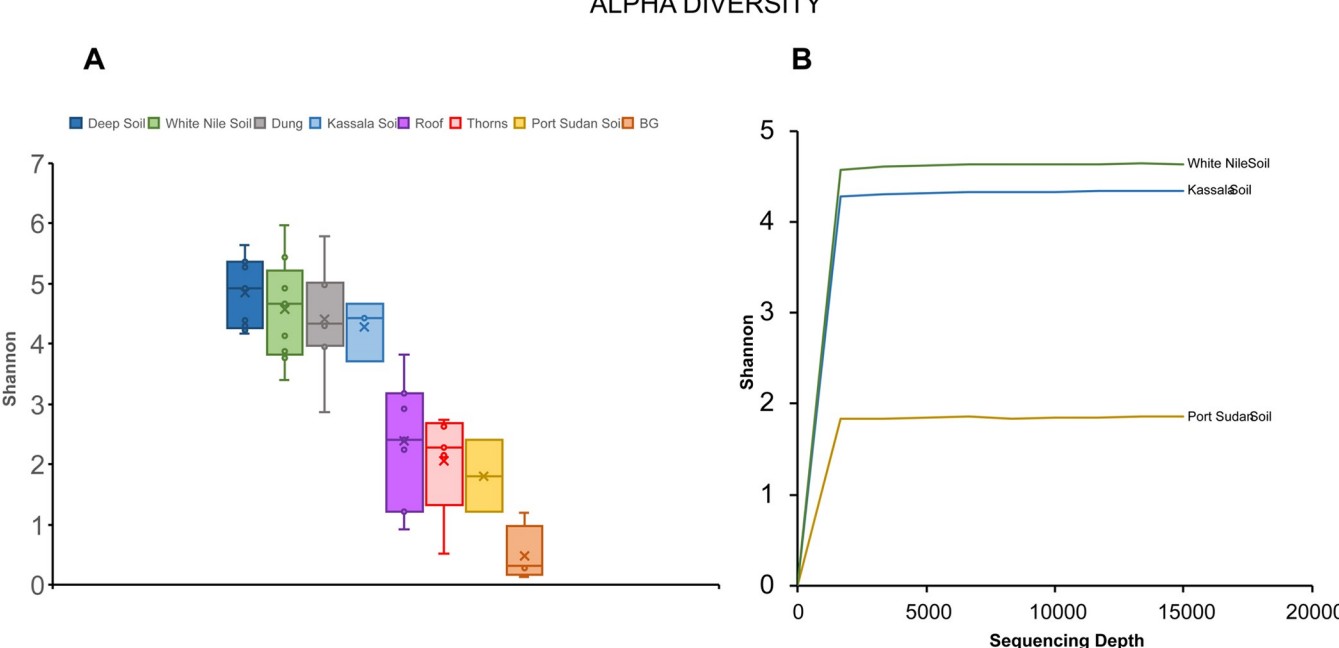

**Fig 2.** Alpha diversity calculated as Shannon index of the samples grouped for type (a): superficial and deep soil, control soil, walls, dung, roof, thorns and black grains and rarefaction curves of endemic and non-endemic samples (b): White Nile Soil is endemic whereas Kassala and Port Sudan is non-endemic.

*M. mycetomatis* represented 10% of all eumycetoma-associated sequences (0.7% of total sequences) across all endemic samples within the study. This places it at the third most abundant eumycetoma-associated species following *Exserohilum rostratum* (*E. rostratum*) (28%) and *Alternaria alternata* (*A. alternata*) (14%). Interestingly, of the above three species only *A. alternata* was found to grow in culture (2%). *M. mycetomatis* was found in all houses, ranging from 0.2% to 6.3% of total sequences (Sheets B-D in S1 Table). No significant difference was

## BETA DIVERSITY

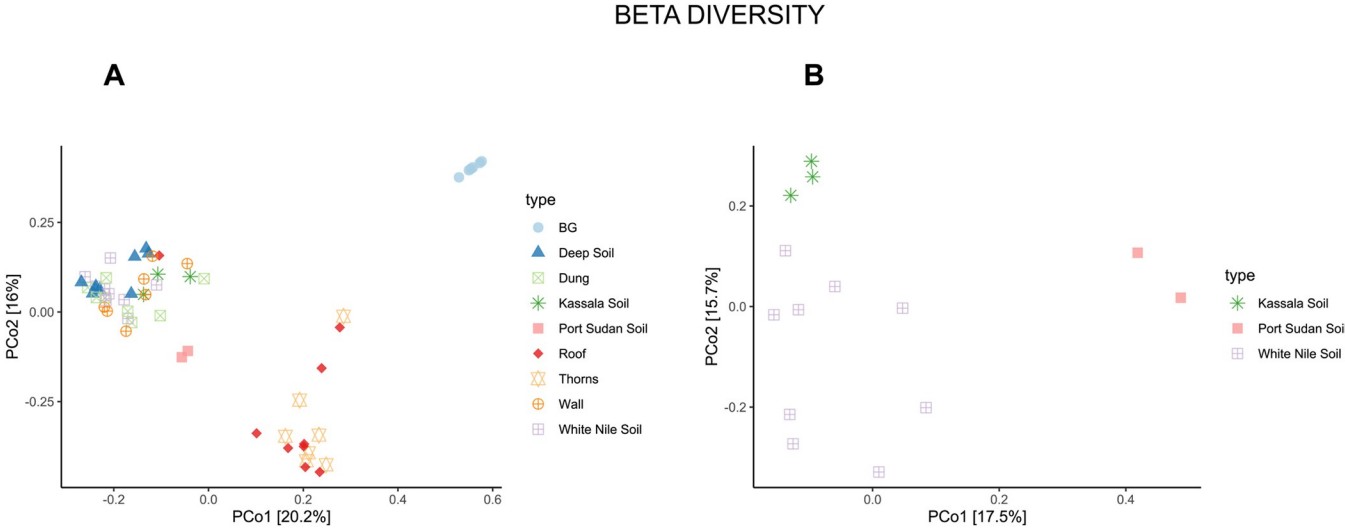

**Fig 3.** Beta diversity calculated as unweighted UniFrac and represented as a Principal Coordinates Analysis (PCoA) of all samples (a) and endemic and non-endemic soil samples (b).

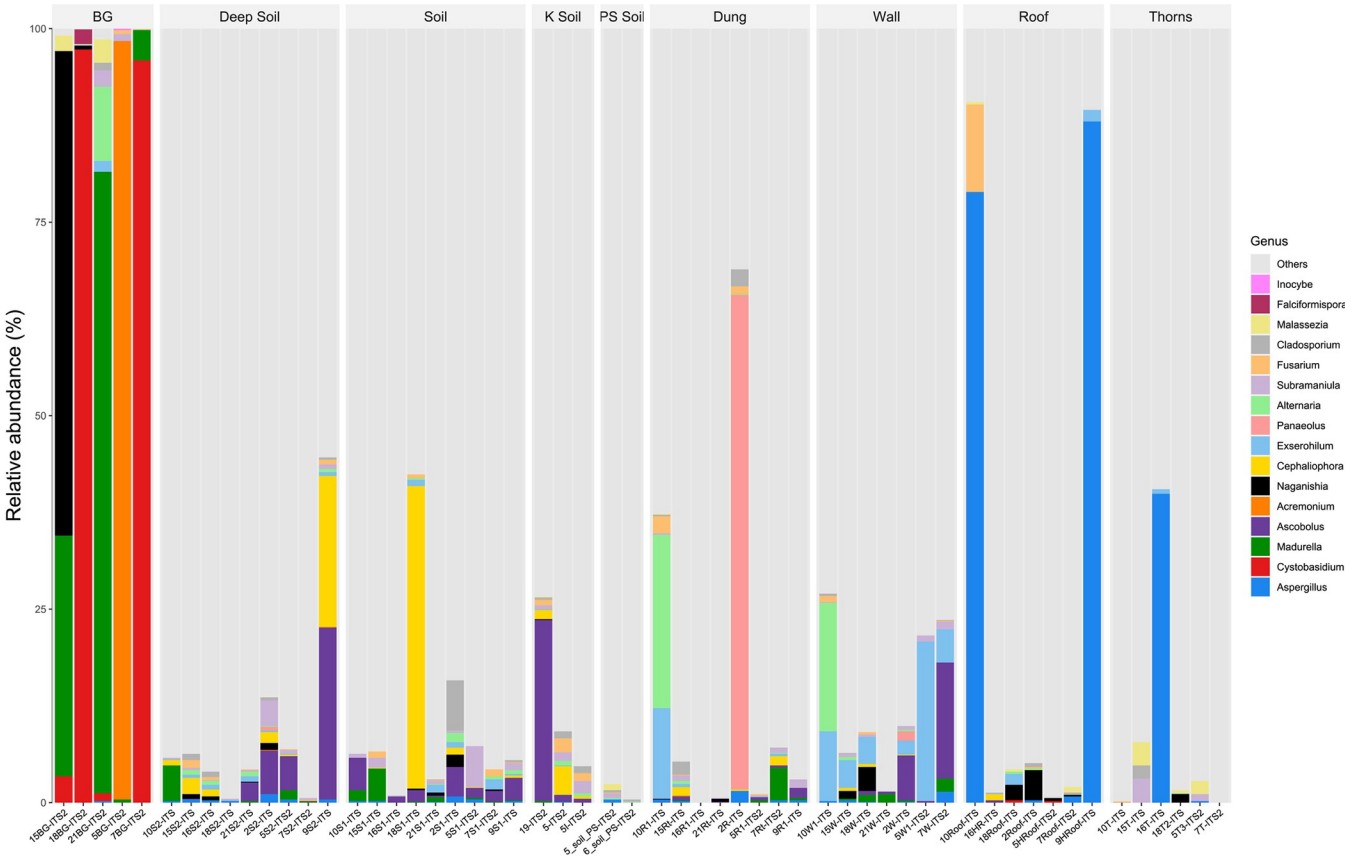

**Fig 4. Taxa abundance in the samples grouped for type, all fungal genera detected in patients black grains samples are shown.**

observed in *M. mycetomatis* presence between the 2 endemic villages (p = 0.352082) (Sheet G in S1 Table). On average, patients affected by eumycetoma-associated with *M. mycetomatis* had more *M. mycetomatis* found within their associated environmental samples when compared with patients with eumycetoma-associated with other fungal genera (4% vs 1%). However, this observed difference did not reach statistical significance ($p$ = 0.0669). As shown in Fig 4, *M. mycetomatis* was not found in roofs and thorns.

### Richness of fungal diversity in soil

Eighteen superficial and deep soil samples from nine houses were sequenced and analysed. Five additional soil samples from not endemic regions were included in the analysis as controls (Table 1 and Fig 1).

ASVs were found in superficial soil samples (234), while 225 were found in deep soil samples with 27 (11.5%) and 25 (11.1%) being from eumycetoma-associated fungi respectively (Sheet B in S1 Table).

*M. mycetomatis* was the most common eumycetoma-associated species demonstrated in both superficial (0.79%) and deep (0.64%) soil samples,. For superficial and deep soil samples this was followed by *Curvularia lunata* (*C. lunata*) (0.61%, 0.5%), *E. rostratum* (0.34%, 0.3%), *Cladosporium* spp. (0.31%, 0.13%), and A. *alternata* (0.3% each), respectively (Sheet C in S1 Table).

*Fusarium solani* (0.02%) and *A. flavus* (0.01%) were also detected in superficial and deep soil samples. *Madurella fahalii* (*M. fahalii*) sequences (0.0001%) were only found in the

superficial soil samples from El Andalos village (house n° 2). Traces of *Acremonium persicum* were also found (0,007%) (Sheet C in S1 Table). Moreover, *Chaetomiaceae* family ASVs were also found in both superficial and deep soil samples (S1B Table). *F. falciforme* and *C. atrobrunneum* were only identified by culture and Sanger sequencing.

Five additional soil samples from Kassala (ASVs = 122) and Port Sudan (ASVs = 82) regions not endemic for eumycetoma were analyzed (Table 1 and Fig 1) and compared to the endemic soil samples. They were less rich in total ASVs and in those associated with eumycetoma (ASVs = 17 and ASVs = 9 respectively) (S1E Table), even if the percentage of eumycetoma-associated fungi in endemic and non-endemic soils did not differ significantly (p = 0.226568) (Sheet G in S1 Table). *M. mycetomatis* was not found in these soil samples, while *M. fahalii* was found in Kassala soil samples representing the 0.07% of the total sequences. Furthermore, *Bipolaris cactivora* (24%) was the most abundant species, followed by *Fusarium spp* (*1.26%*), *A. alternata* (*0.13%*), *E. rostratum* (*0.12%*), *C. lunata* (0.13%) and other eumycetoma species (Sheet E in S1 Table) also found in endemic soil samples.

*Cladiosporium spp* (0.23%), *A. alternata* (0.18%), *A. flavus* (0.03%) and *Fusarium spp* (0.03%) sequences were the only eumycetoma causative species found in Port Sudan soil samples. *E. rostratus* and *Curvularia spp* were statistically more present in El Andalous than in Alsobahi, Wagga and Port Sudan, while in Wagga, *Fusarium spp* and *Bipolaris spp* were statistically more present compared to Alsobahi and El Alandalous, respectively (Sheet G in S1 Table).

Beta diversity represented as PCoA analysis of all samples, including BGs (Fig 3A) and between soil samples from the endemic villages in White Nile compared with the non-endemic soil samples of Wagga and Port Sudan (Fig 3B) is shown. The analysis of all samples shows, as expected, a huge separation between BGs and the other samples (Fig 3A). Roof and thorn samples clustered separately from other samples, including Kassala samples, while Port Sudan soil samples were in the middle (Fig 3A). Moreover, we calculated the beta diversity of only soil samples which proved to be separated, particularly there was a wide distance between the Port Sudan samples and the White Nile samples compared to the Kassala samples.

## Richness of fungal diversity in wall and animal dung samples

Eight animals' dung and seven house wall samples were studied (Table 1). Dung samples included 198 ASVs, 23 of which were associated with eumycetoma, while wall samples included 228 ASVs, 29 of which were associated with eumycetoma.

In the dung samples, *M. mycetomatis* represented the third most abundant eumycetoma causing species (0.68%), preceded by *A. alternata* (2.7%) and *E. rostratum* (1.7%). At genus level *Aspergillus* (6.4%), *Exophiala* (0.9%), *Curvularia* (0.4%) *Cladiosporium* (0.6%) and *Fusarium* (0.3%) were detected. (Sheet D in S1 Table). The most common eumycetoma causing species in the housing walls samples was *E. rostratum* (6%), followed by *A. alternata* (2.3%), *M. mycetomatis* (0.71%), *C. lunata* (0.6%), *A. flavus* (0.02%) and *Cladosporium* genus (0.02%). As in soil samples, ASVs from the *Chaetomiaceae* family were detected (Sheets C and D in S1 Table).

## Richness of fungal diversity in housing roof and thorn samples

Nine roof and 7 thorn samples were studied by metagenomics. 117 ASVs were noted in roof samples with 18 (15%) of them being associated with eumycetoma while 69 ASVs were found in thorns with eight (11.6%) of them being associated with eumycetoma.

In roof samples, the eumycetoma-associated species were *Aspergillus spp* (18.5%). *Fusarium spp* (1.4%), *Exserohilum* (0.9%), *Culvularia spp* (0.2%), *Cladiosporum spp* (0.15%) and *Alternaria* (0.10%). Amongst ASVs with species-level identification, *E. rostratum* (0.9%) and *C. lunata* (0.04%) were detected.

In thorn samples, the only eumycetoma-associated fungus to be identified to species level was *E. rostratum* (3%). However, multiple genus-level ASVs including *Aspergillus* (6.8%) and *Cladosporium* (0.3%) were detected.

Notably, culture was only able to identify *A. niger*, *A. nidulans* and *Fusarium oxysporum* from thorn samples, and *A. nidulans*, *A. niger*, *A. terreus and C. atrobrunneum* from roof samples.

Interestingly, *C. atrobrunneum* was not observed in metagenomics analysis were *Chaetomium* was detected just at genus level.

## Richness of fungal diversity in black grain samples

ITS2 amplicons previously obtained from BG samples (BG5, BG7, BG15, BG18 and BG21) from eumycetoma patients living in Alsobahi (houses #5 and 15) and from El Andalous (houses # 7, 18, and 21) were analysed using the newest version of Qiime2 package ver. 2018.11 (https://doi.org/10.1038/s41587-019-0209-9). The new analysis demonstrated similar results to prior analysis with the Qiime1 package and found both eumycetoma-associated and non-associated species (Sheet F in S1 Table and Fig 4). Notable, the identity of the fungi associated with one patient (BG5 sample) was found to be different on this re-analysis. On first pass the fungus was identified as *Fusarium spp* [18], while on second pass it was called *Acremonium alternatum* (submitted on database in date 26-may-2021_PRJNA732838), *Fusarium spp* and *Acremonium spp* belong to the Hypocreales order, so the ITS2 sequences are very similar. The analysis with Qiime2 is more sophisticated than the older, and also the database is updated, which can explain the difference in the assignment of the species.

Results showed that all eumycetoma-associated fungi detected by metagenomics in BG samples were found in the environmental samples, except *F. senegalensis*, diagnosed in a woman from house # 18 (BG18) and *Acremonium alternatum* detected in a farmer from the house # 5 (BG5). (Sheet F in S1 Table and Fig 4).

Further, three non-eumycetoma-associated fungi, recovered in the same BG samples, were not found in environmental samples, including *Sistotrema adnatum* (BG18), *Inocybe vulpinella*, and *Vishniacozyma* spp. (both in BG5) (Sheet F in S1 Table and Fig 4). Fig 4 visualizes the full range of fungi detected in BG samples.

## Discussion

Mycetoma is a serious and devastating medical and health problem that affects patients, communities, and the health system in endemic regions. It was formally recognised as a neglected tropical disease in 2016 by WHO to improve the management of the patients affected by mycetoma. Presently, there is a massive knowledge gap in mycetoma despite the good efforts to reduce it [22–24]. It is believed that traumatic subcutaneous inoculation of the causative organisms is the possible route of infection in mycetoma, but that is not yet certain. The distribution and the developmental forms of the causative organisms in the soil and environment in endemic regions are also poorly understood [25]. Our understanding of fungi's ecological role in environmental ecosystems in general and in eumycetoma endemic areas is still vague. The data on the ecology of fungi associated with eumycetoma in Sudan are extremely limited [22–26]. Furthermore, the traditional cultural methods for studying fungal diversity in soil and plants have limitations [27].

With this background, this metagenomic study was conducted to gain more insight into the natural habitat of *M. mycetomatis* and other eumycetoma causative agents in two Sudanese endemic villages.

Eleven eumycetoma-associated genera were detected in the environmental samples including *Exserohilum*, *Aspergillus*, *Curvularia*, *Alternaria*, *Madurella*, *Fusarium*, *Cladosporium*,

*Exophiala*, *Microascus*, *Bipolaris*, and *Acremonium*. *M. mycetomatis* was the most abundant eumycetoma-associated agents in the superficial and deep soils, while it represented the third most common species in animals' dungs and house walls' samples by metagenomic. These findings may support the subcutaneous traumatic inoculation infection theory. The soil and the dry feces of animals that live in close contact with people, abound and form the floor of Sudanese rural houses; it is easy to think that these fungi-rich materials (soil, animal dungs and walls) can be inoculated with trauma during daily work, children's play, or other activity as agriculture. Results also suggested that the *Acacia* thorns are probable vectors and do not represent the habitat of *M. mycetomatis*, consistently negative in thorn samples.

The obtained data also showed that fungal eumycetoma causative agents *Acremonium alternatum* (BG5) *and F.* senegalensis (BG18) and additional 3 fungal species not associated to eumycetoma, identified in patients living in house # 5 (N = 1) and house # 18 (N = 2) respectively, were not detected in the environmental samples. It can be extrapolated that eumycetoma infection in these patients was acquired outside the studied villages. That may be supported by the villagers' habit of continuous movement between different villages in the area due to socioeconomic reasons. However, further studies are needed to clarify this.

A previous study predicted the eumycetoma spatial distribution in Sudan, particular in Southeastern villages, along the Nile River, by regression and machine learning techniques [2]. In this pilot study, metagenomic results confirmed the presence of *M. mycetomatis* in these two While Nile endemic villages, suggesting that a relative low percentage of *M. mycetomatis* in the soil and dung samples and consequently in wall samples (ranging from 0.5% to 0.7%) is sufficient to make eumycetoma causing *M. mycetomatis* endemic in these villages.

There was no evidence of *M. mycetomatis* in non-endemic soil, while few sequences of *M. fahalii* were detected in Kassala soil (0.07%). Looking to eumycetoma AVSs diversity, Kassala soil was closer to While Nile soil, with 8 different eumycetoma-associated genera, while just 4 genera associated to eumycetoma were found in Port Sudan soil.

*E. rostratus* and *Curvularia spp* were statistically more present in El Andalous than other endemic and not endemic villages, while *Fusarium spp* and *Bipolaris spp* were statistically more present in one not endemic village (Wagga, Kassala) compared endemic villages (Sheet G in S1 Table).

It appears that soil outside the endemic villages is not endemic for M mycetomatis. Likewise, Port Sudan soil seems not to be endemic for other eumycetoma species with only 1% of the eumycetoma-associated species of all sequences. In contrast, Kassala soil can be considered endemic for other eumycetoma spp, having a percentage of eumycetoma-associated species (23%) even higher than then endemic soils. (17%). These results are in accordance with the distribution of Eumycetoma observed at MRC during 1991–2018 [2].

Metagenomics provided a more comprehensive profile of the fungal species in the soil compared to culture-dependent methods and Sanger sequencing, demonstrating to be a powerful method for both eumycetoma diagnosing and identifying of fungi from environmental samples, even in the absence of growth. There was, however, an inconsistency, *C. atrobrunneum* and *F. falciforme* were only identified by culture and Sanger sequencing and were not detected by metagenomics. It can be attributable to the amplicon metagenomic method that may not be capable of amplifying all cultured fungi due to the biases associated with DNA extraction, PCR amplification and ITS primers. [28]. This shows that the amplicon metagenomic approach based on the ITS2 region of fungal rDNA and the culture method are complementary.

Further interestingly, *Fusarium spp* and *Aspergillus spp*, found in environmental samples from endemic villages (0.4% and 1,7% of ASV sequences, respectively), represented the most recovered species using the culture method.

Although *M. mycetomatis* represented the 0.7% of the total sequences and the 10% of eumycetoma-associated ASVs, it was never isolated from environmental samples by culture. This is in line with previous reports [29].

The exact explanation for this finding is not clear.

The abundant fungal species in the environmental samples may explain the *M. mycetomatis* growth inhibition on culture media. That may be due to the overgrowth of other microorganisms or the presence of antagonistic fungi or bacteria, or inhibitory compounds induced from fungal interactions, as previously shown in co-culturing experiments [30,31].

The fact that *M. mycetomatis* usually in vivo takes over the other fungi could be due to its ability to produce melanin and black grains to escape the various host immune responses. Another possible explanation for the difficulty of *M. mycetomatis* cultivation in vitro could be its ability to produce melanin and black grains. While this ability to escape immune responses may aid in producing such chronic infections in vivo, in vitro this may produce an effect that inhibits growth and sporulation like a bacterial biofilm or endospore. This idea could further be supported by the observation of hyaline molds and non-melanin containing fungi being more frequently isolated from both immunocompromised and malnourished patients than their dematiaceous counterparts.

While the conclusions in this study may be limited by its small sample size it does offer some suggestions for reducing the disease burden felt by those living in endemic and often impoverished areas. These include using non-endemic soil, avoiding animal dung in housing construction, and using proper footwear and gloves in agricultural and other activities that may facilitate traumatic inoculation.

Despite the fact that metagenomics proved to be extremely useful in assaying environmental and clinical samples in order to better understand the relationship between environmental fungi and eumycetoma pathogens, the implementation of such techniques in developing countries is still restricted by financial limitations.

Given the significant results of the study, this research has been recently funded and will be extended to 100 eumycetoma patients' houses, including clinical samples and different environmental samples from endemic and non-endemic villages.

## Supporting information

**S1 Table.** Sheets: A: Cultural and Sanger results; B: ASV in samples from White Nile (sequences); C: ASV in samples grouped for type and houses (%); D: Table of fungal species and genera associated with eumycetoma, found in the different sample types (%); E: ASV in soil samples from the non-endemic region (%); F: ASV in black grains samples (%). G: correlation between the eumycetoma associated species and the different sites.
(XLSX)

## Acknowledgments

We thank Gabriele Carenti and Emanuela Sias for the technical assistance for the preparation of Fig 1.

## Author Contributions

**Conceptualization:** Antonella Santona, Emmanuel Edwar Siddig, Maura Fiamma, Salvatore Rubino, Ahmed Hassan Fahal.

**Data curation:** Antonella Santona, Najwa A. Mhmoud, Emmanuel Edwar Siddig, Massimo Deligios, Maura Fiamma, Sahar Mubarak Bakhiet, Salvatore Rubino, Ahmed Hassan Fahal.

**Formal analysis:** Antonella Santona, Najwa A. Mhmoud, Emmanuel Edwar Siddig, Maura Fiamma, Salvatore Rubino, Ahmed Hassan Fahal.

**Funding acquisition:** Antonella Santona, Salvatore Rubino, Ahmed Hassan Fahal.

**Investigation:** Antonella Santona, Najwa A. Mhmoud, Emmanuel Edwar Siddig, Massimo Deligios, Maura Fiamma, Bianca Paglietti, Sahar Mubarak Bakhiet, Salvatore Rubino, Ahmed Hassan Fahal.

**Methodology:** Antonella Santona, Emmanuel Edwar Siddig, Massimo Deligios, Maura Fiamma, Bianca Paglietti, Sahar Mubarak Bakhiet, Salvatore Rubino, Ahmed Hassan Fahal.

**Project administration:** Antonella Santona, Salvatore Rubino, Ahmed Hassan Fahal.

**Resources:** Antonella Santona, Massimo Deligios, Salvatore Rubino, Ahmed Hassan Fahal.

**Software:** Antonella Santona, Emmanuel Edwar Siddig, Massimo Deligios, Maura Fiamma, Salvatore Rubino, Ahmed Hassan Fahal.

**Supervision:** Antonella Santona, Maura Fiamma, Salvatore Rubino, Ahmed Hassan Fahal.

**Validation:** Antonella Santona, Najwa A. Mhmoud, Emmanuel Edwar Siddig, Massimo Deligios, Maura Fiamma, Bianca Paglietti, Sahar Mubarak Bakhiet, Salvatore Rubino, Ahmed Hassan Fahal.

**Visualization:** Antonella Santona, Najwa A. Mhmoud, Emmanuel Edwar Siddig, Massimo Deligios, Maura Fiamma, Bianca Paglietti, Salvatore Rubino, Ahmed Hassan Fahal.

**Writing – original draft:** Antonella Santona, Najwa A. Mhmoud, Emmanuel Edwar Siddig, Massimo Deligios, Maura Fiamma, Bianca Paglietti, Sahar Mubarak Bakhiet, Salvatore Rubino, Ahmed Hassan Fahal.

**Writing – review & editing:** Antonella Santona, Emmanuel Edwar Siddig, Massimo Deligios, Maura Fiamma, Bianca Paglietti, Salvatore Rubino, Ahmed Hassan Fahal.

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
