## [Decision Letter · Decision Letter 0]

9 May 2022

Dear Dr. Siddig,

Thank you very much for submitting your manuscript "Metagenomics detection of eumycetoma causative agents within eumycetoma patient’s household environment in two Sudanese endemic villages, in White Nile State." for consideration at PLOS Neglected Tropical Diseases. As with all papers reviewed by the journal, your manuscript was reviewed by members of the editorial board and by several independent reviewers. In light of the reviews (below this email), we would like to invite the resubmission of a significantly-revised version that takes into account the reviewers' comments. 

We cannot make any decision about publication until we have seen the revised manuscript and your response to the reviewers' comments. Your revised manuscript is also likely to be sent to reviewers for further evaluation.

Sincerely,

Angel Gonzalez, Ph.D.

Associate Editor

Todd Reynolds

Deputy Editor

Reviewer's Responses to Questions

**Key Review Criteria Required for Acceptance?**

**Methods**

-Are the objectives of the study clearly articulated with a clear testable hypothesis stated?

-Is the study design appropriate to address the stated objectives?

-Is the population clearly described and appropriate for the hypothesis being tested?

-Is the sample size sufficient to ensure adequate power to address the hypothesis being tested?

-Were correct statistical analysis used to support conclusions?

-Are there concerns about ethical or regulatory requirements being met?

Reviewer #1: The manuscript metagenomics detection of eumycetoma causative agents is an interesting study that adds to the existing literature concerning possible environmental niches and routes of acquisition of this devastating neglected tropical disease.

Although the study size (sample numbers) is relatively modest, this is comprehensible given the laborious nature of the approaches used .

The methods employed are appropriate for the study aim, and the experiments appear to have been well-conducted.

On the whole, the data support the conclusions drawn by the authors, and are presented in a clear fashion.

My only significant criticism is of the English usage/grammar. The entire manuscript would benefit form an extensive revision by a native English speaker.

Reviewer #2: Yes to all the above except as noted in attachment

Reviewer #3: The article “Metagenomics detection of eumycetoma causative agents within eumycetoma patient’s household environment in two Sudanese endemic villages, in White Nile State” by Antonella Santona et al. aims to provide the epidemiological information about the source of causative agents for mycetoma. Using the metagenomics approach, the previously unidentified causative species might be easier identified when compared to the detection by cultured dependent methods. Assuming that this is the study’s novelty, however, this point was not clearly stated in the introduction part of the article.

For the method, the study was designed based on a previously published study site, so the sample size might be too small to answer the question at large and might not be able to apply in the other endemic area, which leads to the lack of the external validity of the study. However, the information might be used in endemic areas with similar environmental factors. The authors should discuss this limitation more.

The method of metagenomics analyses is well described and is enough to answer the research questions. However, several analyses are possible to be added by using the existing data set to create a more sophisticated article, for example, the rarefaction analyses of each sample, the comparison of the phylogenetic diversity (PD) between each sample, especially from endemic vs. non-endemic areas, or other kinds of alpha diversity such as Chao1, etc. Please consider adding these analyses.

One additional weak point of this study is the lack of data about the abundance of the mycetoma-associated taxon in other type of samples in the non-endemics area except soil (e.g., roof, water). So, it is hard to conclude that the present of the mycetoma-associated taxon in each type of sample in endemics areas might are the indeed source of the infection. The authors should consider adding this part of experiment or discuss about this limitation.

The other concern of the method is the availability of the sequencing data, which the authors failed to demonstrate or still do not deposit on any repository (for example, GenBank).

**Results**

-Does the analysis presented match the analysis plan?

-Are the results clearly and completely presented?

-Are the figures (Tables, Images) of sufficient quality for clarity?

Reviewer #1: The manuscript metagenomics detection of eumycetoma causative agents is an interesting study that adds to the existing literature concerning possible environmental niches and routes of acquisition of this devastating neglected tropical disease.

Although the study size (sample numbers) is relatively modest, this is comprehensible given the laborious nature of the approaches used .

The methods employed are appropriate for the study aim, and the experiments appear to have been well-conducted.

On the whole, the data support the conclusions drawn by the authors, and are presented in a clear fashion.

My only significant criticism is of the English usage/grammar. The entire manuscript would benefit form an extensive revision by a native English speaker.

Reviewer #2: Yes to all the above except as noted in attachment

Reviewer #3: Most of the result are clearly presented according to analysis plan and the pipeline of metagenomics analysis. However the results is not clearly presented and several important imformation is still lacking, for example most of the results are presented in the abundance of the mycetoma-associated taxon in each sample pattern. However, the abundance of the mycetoma-associated taxon in the endemic area compared to non-endemic regions should be included and shown by using a statistical calculation to prove that the increase in the percentage of the mycetoma-associated taxon is genuinely associated with the endemicity of the disease. Most of the study is still lacking in these essential results (as mentioned in methods), or some sample was already done (soil) (for example, lines 318 – 320), but the statistical analysis has still lacked. Therefore, an additional analysis of this dataset is needed.

In addition to the analysis, the table and figures also should be revised

In table 1, the name of sample types should be revised (what are the meaning of S1, S2, W,…?).

In figure legends, please describe the meaning of abbreviations such as K soil and PS soil.

Please considers adding color to the beta-diversity PCoA plot (Figure 3) and add a sub-figure that extracts only the information about soil, especially to demonstrate the difference between endemic and non-endemics areas.

**Conclusions**

-Are the conclusions supported by the data presented?

-Are the limitations of analysis clearly described?

-Do the authors discuss how these data can be helpful to advance our understanding of the topic under study?

-Is public health relevance addressed?

Reviewer #1: The manuscript metagenomics detection of eumycetoma causative agents is an interesting study that adds to the existing literature concerning possible environmental niches and routes of acquisition of this devastating neglected tropical disease.

Although the study size (sample numbers) is relatively modest, this is comprehensible given the laborious nature of the approaches used .

The methods employed are appropriate for the study aim, and the experiments appear to have been well-conducted.

On the whole, the data support the conclusions drawn by the authors, and are presented in a clear fashion.

My only significant criticism is of the English usage/grammar. The entire manuscript would benefit form an extensive revision by a native English speaker.

Reviewer #2: Yes to all the above except as noted in attachment

Reviewer #3: The public health relevance was clearly addressed in the discussion and conclusion part. But the limitation of the study should be more focused than the small sample size. The discussion has required revision; for example, what is the importance of the presence of each mycetoma-associated taxon or the possible route of infection from the source by citing the previously published data or related to the local activity which not clearly illustrated for a general audience.

Lastly, the conclusion and suggestion of the study, such as changing house building material, might be a slightly over conclusion because many infection risk factors are required to identify. So the conclusion might be that “the soil in the endemic area might be the source of infection” would be more suitable.

**Editorial and Data Presentation Modifications?**

Reviewer #1: Extensive correction/revision of the English would benefit this manuscript. It is not particularly badly written, it is more that the English is clumsy in many places, which detracts from the read.

Reviewer #2: See attachment. While most changes needed are editorial in nature they, along with a couple informational and conceptual questions per the attachment, are important enough to impact clarity and reader understanding.

Reviewer #3: (No Response)

**Summary and General Comments**

Reviewer #1: The manuscript metagenomics detection of eumycetoma causative agents is an interesting study that adds to the existing literature concerning possible environmental niches and routes of acquisition of this devastating neglected tropical disease.

Although the study size (sample numbers) is relatively modest, this is comprehensible given the laborious nature of the approaches used .

The methods employed are appropriate for the study aim, and the experiments appear to have been well-conducted.

On the whole, the data support the conclusions drawn by the authors, and are presented in a clear fashion.

My only significant criticism is of the English usage/grammar. The entire manuscript would benefit form an extensive revision by a native English speaker.

Reviewer #2: See attachment.

Reviewer #3: Overall, even though the study still has a big room for improvement, but the information from this study might be an important key for the prevention and control of the disease. The reanalyses of the data is needed. 

Moreover, there are a major concern about the ethical issues in this study for example the use of isolated specimen from human without state about the IRB approval, and the availability of sequencing data.

PLOS authors have the option to publish the peer review history of their article (what does this mean?). If published, this will include your full peer review and any attached files.

Reviewer #1: No

Reviewer #2: No

Reviewer #3: Yes: Apisit Chaidee
---

## [Editor Report · Decision Letter 1]

14 Jul 2022

Dear Dr. Sidding,

We are pleased to inform you that your manuscript 'Metagenomic detection of eumycetoma causative agents   from households of patients residing in two Sudanese endemic villages in White Nile State.' has been provisionally accepted for publication in PLOS Neglected Tropical Diseases.

Best regards,

Angel Gonzalez, Ph.D.

Academic Editor

Todd Reynolds

Section Editor

---

## [Editor Report · Acceptance letter]

22 Aug 2022

Dear Dr Siddig,

We are delighted to inform you that your manuscript, "Metagenomic detection of eumycetoma causative agents   from households of patients residing in two Sudanese endemic villages in White Nile State.," has been formally accepted for publication in PLOS Neglected Tropical Diseases.

Best regards,

Shaden Kamhawi

co-Editor-in-Chief

Paul Brindley

co-Editor-in-Chief
